# Mutual Inductance and Load Identification of LCC-S IPT System Considering Equivalent Inductance of Rectifier Load

**Haomin Shen, Xiaona Wang \*, Pan Sun \*, Lei Wang and Yan Liang**

School of Electrical Engineering, Naval University of Engineering, Wuhan 430030, China;
min22nim@126.com (H.S.); 15198243251@163.com (L.W.); hgliangyan@126.com (Y.L.)
\* Correspondence: wxnsd@163.com (X.W.); chinasunpan@163.com (P.S.); Tel.: +86-138-7116-7295 (X.W.);
+86-159-7222-6502 (P.S.)

**Abstract:** The variation of mutual inductance and load parameters will affect the transmission power and efficiency of the inductive power transfer (IPT) system. The identification of mutual inductance and load parameters is an essential part of establishing a stable and reliable IPT system. This paper presents a joint identification method of load and mutual inductance for the LCC-S IPT system, which does not require the establishment of primary and secondary communication and related control. Firstly, the resistance-inductance characteristics of the equivalent load of the rectifier are analyzed by simulation, and then the rectifier and system load are equivalent to the circuit model of resistance and inductance in series. Secondly, the characteristics of the reflected impedance are analyzed, and the functional relationship between the transmitter impedance and the rectifier impedance is established by using the ratio of the real part to the imaginary part of the reflected impedance, which realizes the decoupling of the load and the mutual inductance. Thirdly, the functional relationship between the equivalent impedance of the rectifier and the load resistance of the system is obtained by data fitting. Then, the equations of the above two functional relationships are combined. By measuring the voltage of the parallel compensation capacitor at the transmitting side, the current of the transmitting coil and the phase difference between the two, the battery load can be solved first, and then the mutual inductance can be calculated, so that the high-precision identification of the load and mutual inductance can be realized. Finally, an experimental platform of the LCC-S IPT system is built for experimental verification. The experimental results show that the maximum identification errors of mutual inductance and load are 5.20% and 5.53%, respectively, which proves that the proposed identification method can achieve high precision identification.

**Keywords:** inductive power transfer (IPT); mutual inductance parameter identification; load parameter identification; rectifier equivalent load

## 1. Introduction

With the development of science and technology, inductive power transfer (IPT) technology has attracted more and more attention. Compared with the traditional charging method, this technology is safe, reliable, flexible and convenient, and has strong adaptability [1–6]. As a new type of power supply, it is widely used in electric vehicle charging [7], electronic products charging [8], biomedical [9] and underwater vehicle fields [10], as well as other fields. In addition, with the continuous development of the IPT system, its application scenarios are also being further explored and expanded.

In the practical application of the IPT system, the change of the relative position between the transmitter and the receiver will lead to the change of the mutual inductance of the coupler. Moreover, with the change of the charging state of the charging device itself or the switching of different devices, the load of the system is also in the process of dynamic change. The change of mutual inductance and load of the system will cause the change of the system reflection impedance, which will affect the transmission power and

transmission efficiency of the system [11]. Therefore, in order to establish a more efficient and reliable IPT system, it is necessary to identify the parameter information of mutual inductance and load, and then guide the control of the system. The most critical problem is the identification of mutual inductance and load parameters [11,12].

At present, some scholars have carried out research on the identification of the load and mutual inductance parameters of the IPT system [13–19]. Ref. [13] establishes the steady-state circuit model of the SS type IPT system, uses the genetic algorithm to obtain the optimal solution of the load and then obtains the mutual inductance value. Ref. [14] makes the system work in two operating modes by switching capacitors, and then establishes a mathematical model based on these two modes to identify the load and mutual inductance parameters of the SS-type IPT system. Refs. [15–17] establish the energy supply, storage and dissipation functions and energy balance equation of the system based on the method of energy analysis. Then, they propose a load parameter identification method for the SS-type IPT system by analyzing the reflected impedance. In Ref. [18], a parameter identification method based on the PyTorch neural network is proposed for the LCC-S IPT system. This method transforms the parameter identification problem into a deep learning nonlinear fitting problem by training the neural network model. In Ref. [19], a load and mutual inductance identification method based on the improved grey wolf optimization algorithm is proposed, aiming at the LCC-S type IPT system. This method takes the real-time current mathematical model of the transmitting side as the identification model, and aims at minimizing the real-time current error of the transmitting side. Then, the parameter identification problem is transformed into an optimization problem to identify the mutual inductance and load in real time.

The traditional parameter identification methods need to add additional communication equipment at the receiver, which will increase the complexity of the system [20]. Furthermore, in the existing research on the mutual inductance and load parameter identification method of the IPT system, there are mainly the following problems:

1.  There are many studies on the parameter identification of the SS type IPT system, but few on the LCC—S type IPT system.
2.  Some parameter identification methods can only identify a single parameter.
3.  The nonlinear characteristic of a uncontrolled rectifier circuit is not considered in the traditional parameter identification research [21]. If the rectifier and its back-end are only regarded as pure resistors, the accuracy of the system model will be reduced, thus affecting the accuracy of parameter identification.

In order to achieve a high-precision identification of mutual inductance and load parameters of the IPT system, this paper proposes a joint identification method of mutual inductance and load parameters for the LCC-S type IPT system. The rest of the article is arranged as follows: Section 2 analyzes the working principle of the LCC-S IPT system. Section 3 analyzes the characteristics of the reflection impedance to realize the decoupling of the mutual inductance and load. Then, a joint identification method of mutual inductance and load parameters is proposed, and the equivalent impedance of the rectifier is modeled by the method of data fitting. Section 4 provides the simulation and experimental verification analysis. Finally, the full article is summarized in Section 5.

## 2. Basic Analysis of LCC-S IPT System

Compared with the low-order compensation topology, the transmitter adopts the LCC topology, which can achieve a constant voltage or constant current output by configuring the parameters of the compensation network and setting the operating frequency. This is in line with the practical application requirements of the dynamic power supply, and it is easy to realize zero voltage switching (ZVS) [22] to reduce switching loss. The receiving side adopts an S-type topology, which is conducive to reducing the number of components and reducing the weight of the device. It has received extensive attention in the field of electric vehicle charging. Therefore, this paper takes the LCC-S IPT system as an example to study.

The circuit topology of the LCC-S IPT system is shown in Figure 1. It is composed of DC power supply, full-bridge inverter circuit, coils, compensation network, uncontrolled rectifier circuit and load. $L_p$, $L_s$ and $M$ are the self-inductance of the transmitting coil, the self-inductance of the receiving coil and the mutual inductance between the two coils, respectively. $L_r$, $C_r$ and $C_p$ are the series compensation inductance, parallel compensation capacitance and series compensation capacitance of the primary side compensation network, respectively. $C_s$ is the secondary side series compensation capacitor. $R_p$ and $R_s$ denote the equivalent internal resistance of the transmitting and receiving coils. $U_b$ and $I_b$ are the charging voltage and charging current of the load $R_L$. The working principle of the system is as follows: The DC voltage source $U_{dc}$ provides the power input of the whole system. Then, the switches of $G_1{\sim}G_4$ constitute a high-frequency inverter, which converts the DC voltage into a high-frequency square wave voltage and sends it to the original side compensation network. After the filtering and reactive power compensation of the original side compensation network, the high-frequency alternating current is transmitted to the secondary side through the induction coil. Diodes $D_1{\sim}D_4$ and the filter capacitor $C_o$ constitute the rectifier filter circuit. They rectify the high frequency alternating current on the secondary side to the direct current, and then charge the system load.

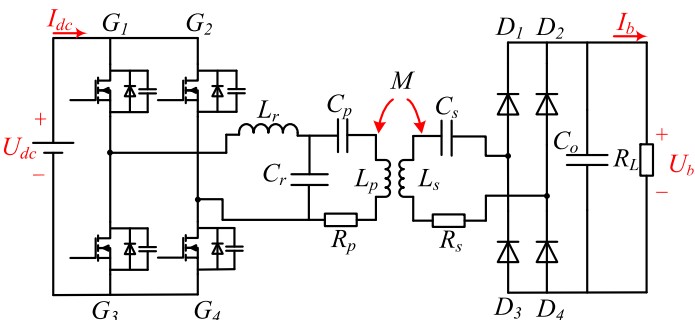

**Figure 1.** Circuit structure of LCC-S IPT system.

By simplifying the system of Figure 1, the equivalent circuit diagram of the system shown in Figure 2 can be obtained. $u_{inv}$ and $i_{inv}$ are the inverter output voltage and current, respectively. $u_{rec}$ and $i_{rec}$ are the input voltage and current of equivalent load, respectively. $u_p$ is the voltage on the parallel compensation capacitor $C_r$ at the transmitter. $i_p$ is the current of the transmitting coil. $R_{eq}$ is the equivalent load resistance of the rectifier module and $\omega$ is the angular frequency of the system.

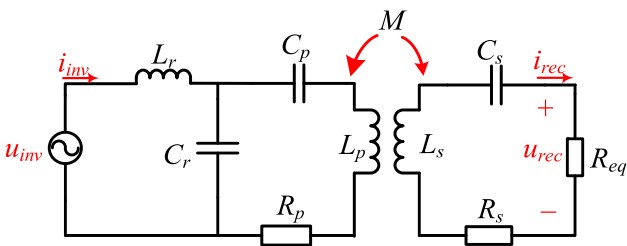

**Figure 2.** Equivalent Circuit of LCC-S IPT System.

In the IPT system, the coupling coefficient is usually used to characterize the tightness of the coupling between the transmitting coil and the receiving coil. The calculation method of the coupling coefficient $e$ is as follows:

$$e = \frac{M}{\sqrt{L_p L_s}} \implies M = e\sqrt{L_p L_s} \tag{1}$$

As observed from (1), the mutual inductance $M$ depends on the coupling coefficient $e$ and the self-inductance of the coils.

Based on Kirchhoff's voltage law, the depiction of the LCC-S system can be obtained as follows:

$$\begin{cases} j\omega L_r \dot{I}_{inv} + \dot{I}_p \left[ j\left( \omega L_p - \frac{1}{\omega C_p} \right) + R_p \right] - j\omega M \dot{I}_{rec} = \dot{U}_{inv} \\ j\omega L_r \dot{I}_{inv} + \left( \dot{I}_{inv} - \dot{I}_p \right) \frac{1}{j\omega C_r} = \dot{U}_{inv} \\ \dot{I}_{rec} \left[ j\left( \omega L_s - \frac{1}{\omega C_s} \right) + R_s + R_{eq} \right] - j\omega M \dot{I}_p = 0 \end{cases} \tag{2}$$

In order to improve the energy transmission efficiency of the system, the operating angular frequency $\omega$ of the system is generally made close to the natural resonant frequency of the circuit [23–25]. Therefore, the parameters of the coupler and compensation networks should satisfy the following relationship:

$$\begin{cases} \omega L_r = \frac{1}{\omega C_r} \\ \omega L_s = \frac{1}{\omega C_s} \\ \omega \left( L_p - L_r \right) = \frac{1}{\omega C_p} \end{cases} \tag{3}$$

Then, the current of each loop under the resonance condition can be determined in (4).

$$\begin{cases} \dot{I}_{inv} = \frac{C_r \left[ \left( R_s + R_{eq} \right) R_p + \omega^2 M^2 \right]}{L_r \left( R_s + R_{eq} \right)} \dot{U}_{inv} \\ \dot{I}_p = -j\omega C_r \dot{U}_{inv} \\ \dot{I}_{rec} = \frac{M}{L_r \left( R_s + R_{eq} \right)} \dot{U}_{inv} \end{cases} \tag{4}$$

Therefore, the voltage gain of the LCC-S system can be determined in (5), and the current gain of the LCC-S system can be determined in (6).

$$G_v = \left| \frac{\dot{U}_{rec}}{\dot{U}_{inv}} \right| = \frac{M R_{eq}}{L_r \left( R_s + R_{eq} \right)} = \frac{M}{L_r} \tag{5}$$

$$G_i = \left| \frac{\dot{I}_{rec}}{\dot{I}_{inv}} \right| = \frac{M}{C_r \left[ \left( R_s + R_{eq} \right) R_p + \omega^2 M^2 \right]} \tag{6}$$

It can be observed from (4) that the expression of the current of the transmitting coil $i_p$ on the primary side is independent of the load $R_L$, so the primary side of the LCC-S type IPT system can achieve a constant current output when the system is resonant. Similarly, it can be observed from (5) that the output voltage of the secondary side $u_{rec}$ is independent of the load $R_L$, so the secondary side of the LCC-S type IPT system has the characteristics of constant voltage output. Moreover, the voltage gain is only related to the topology network parameters and the mutual inductance of the coil, and is not affected by load changes. Therefore, it is suitable for dynamic charging and other application environments.

## 3. Joint Identification Method of Mutual Inductance and Load Parameters

This section may be divided by subheadings. It should provide a concise and precise description of the experimental results and their interpretation, as well as the experimental conclusions that can be drawn.

### 3.1. Rectifier Equivalent Load Analysis of LCC-S Type IPT System

In addition to system components, the performance of an IPT system is also affected by load characteristics. The diode has the advantages of simple structure, high stability, and no additional control, therefore, the receiver of the IPT system usually adopts a full-bridge uncontrolled rectifier circuit. The rectifier and the back-end circuit are usually equivalent to a resistor $R_{eq}$, as shown in Figure 3 [15,16]. The relationship between the $R_{eq}$ and $R_L$ can be expressed by (7).

$$R_{eq} = \frac{8}{\pi^2} R_L \tag{7}$$

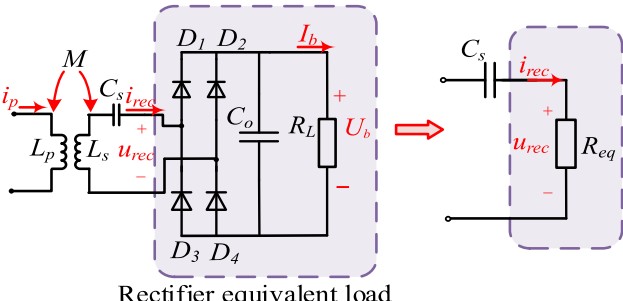

**Figure 3.** Traditional rectifier equivalent load circuit model.

In fact, equivalent load model of rectifier also contains inductance components, so it is not accurate to equate it to a pure resistor. The rectifier and its back end should be represented as a series circuit consisting of an equivalent input resistance and an equivalent input inductance [17]. In addition, based on the double-sided LCC system, the quantitative relationship between the equivalent load resistance and the equivalent input impedance of rectifier is deduced by analyzing the relationship between the voltage on the parallel compensation capacitor at the receiver, rectifier input voltage and rectifier input current. Therefore, the parameter identification with higher precision is realized for the double-sided LCC system.

The input voltage and input current waveforms of the rectifier in the LCC-S type IPT system are shown in Figure 4. Here, $u_{rec}$ and $i_{rec}$ are the input voltage and current of the rectifier, respectively; $u_{rec\_f}$ and $i_{rec\_f}$ are the fundamental components of the input voltage and current of the rectifier, respectively. It can be observed from Figure 4 that the input current waveform of the rectifier $i_{rec}$ is not a standard sine wave, and there is a certain distortion. Moreover, compared with the fundamental component $i_{rec\_f}$, its amplitude is to the right. This indicates that the equivalent load of the rectifier is not purely resistive. In addition, the phase of the fundamental wave of the input voltage of the rectifier is ahead of the phase of the fundamental wave of the input current of the rectifier, which can also indicate that the equivalent load of the rectifier is inductive. Therefore, it is not accurate that the rectifier and the back end are only equivalent to pure resistance in the process of parameter identification, as shown in the Figure 3.

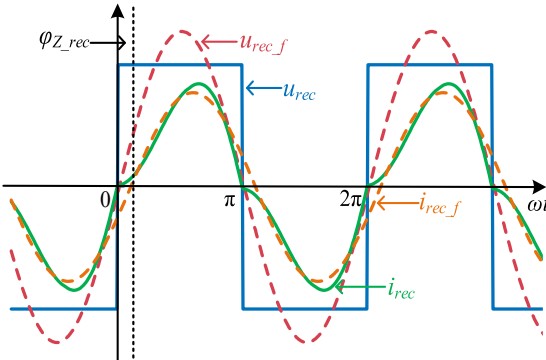

**Figure 4.** Schematic of rectifier input voltage $u_{rec}$ and current $i_{rec}$.

In order to verify the inductive component of the rectifier equivalent load module, a model of an IPT system as depicted in Figure 1 is built in the Matlab/Simulink environment and sets the variation range of the system load $R_L$ to 30~80 Ω. The phase angle of the input voltage $u_{rec}$ and the input current $i_{rec}$ of the rectifier equivalent load module is obtained through Fast Fourier Transform (FFT), and then the impedance angle of the rectifier equivalent load $\varphi_{Z\_rec}$ under different load conditions is calculated. It can be observed from Figure 5 that in the range of load $R_L$, the impedance angle of the equivalent load of the rectifier is greater than zero, which further indicates that the equivalent load

of the rectifier has resistance-inductance characteristics. Moreover, as the value of system load $R_L$ increases, the impedance angle of the equivalent load of the rectifier $\varphi_{Z\_rec}$ also increases. It shows that the system load $R_L$ will affect the inductance of the equivalent load of the rectifier.

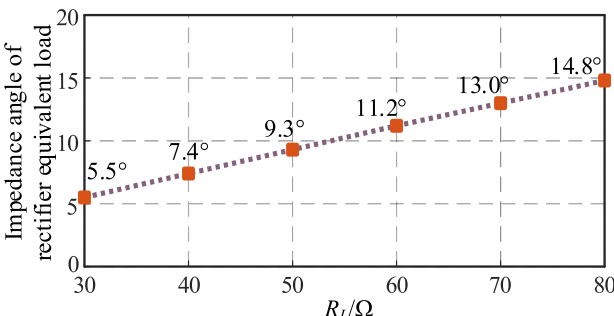

**Figure 5.** Impedance angle of rectifier load of LCC-S topology under different load conditions.

The LCC-S type IPT system is different from the double-sided LCC system because there is no parallel compensation capacitor at the front end of the rectifier. The quantitative relationship between the system load resistance and the equivalent impedance of the rectifier derived in reference [17] cannot be used. Therefore, in the process of the parameter identification of the LCC-S type IPT system, in order to ensure the identification accuracy, the inductance component of the equivalent load module of the rectifier must be considered.

### 3.2. Joint Identification of Mutual Inductance and Load Parameters

The equivalent load module of the rectifier is equivalent to a circuit model in which a resistor and an inductor are connected in series, as shown in Figure 6. $R_e$ is the equivalent input resistance, and $L_e$ is the equivalent input inductance. Define $X_e$ as the equivalent reactance, where $X_e = \omega L_e$.

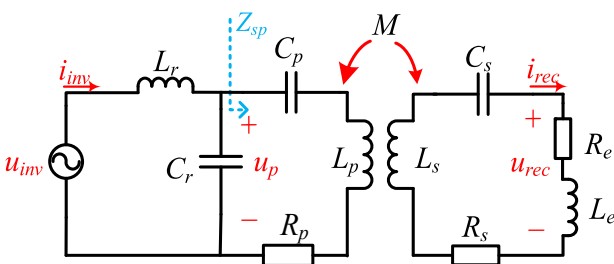

**Figure 6.** Equivalent circuit model of LCC-S type IPT system considering equivalent inductance of rectifier.

Seen from Figure 5, as the load resistance increases, the impedance angle of the equivalent impedance of the rectifier $\varphi_{Z\_rec}$ increases monotonically. Although the variation law of the equivalent input impedance of the rectifier is uncertain, it can be reasonably inferred that there is a certain functional relationship between the equivalent input impedance of the rectifier and system load. This means that $R_e$ and $X_e$ can be represented by $R_L$, as shown in (8).

$$\begin{cases} R_e = f(R_L) \\ X_e = g(R_L) \end{cases} \tag{8}$$

According to the circuit principle, the equivalent impedance of the receiver can be derived as:

$$Z_s = R_s + f(R_L) + j\left[\omega L_s - \frac{1}{\omega C_s} + g(R_L)\right] \tag{9}$$

Then, the reflected impedance equivalent to the transmitter can be derived as:

$$Z_r = \frac{\omega^2 M^2}{Z_s} \tag{10}$$

Taking (9) into (10) and separating the real part and imaginary part of $Z_r$, (11) can be obtained.

$$Z_r = real(Z_r) + imag(Z_r)j \tag{11}$$

where

$$\begin{cases} real(Z_r) = \frac{\omega^2 M^2}{[R_s + f(R_L)]^2 + [\omega L_s - \frac{1}{\omega C_s} + g(R_L)]^2} [R_s + f(R_L)] \\ imag(Z_r) = \frac{-\omega^2 M^2}{[R_s + f(R_L)]^2 + [\omega L_s - \frac{1}{\omega C_s} + g(R_L)]^2} \left[\omega L_s - \frac{1}{\omega C_s} + g(R_L)\right] \end{cases}$$

The coefficient $\alpha$ is defined as the ratio of the real part to the imaginary part of $Z_r$, as shown in (12). It can be observed that $\alpha$ is related to $R_L$ and has nothing to do with the mutual inductance of the coil $M$, thus realizing the decoupling of $R_L$ and $M$.

$$\alpha = \frac{real(Z_r)}{imag(Z_r)} = -\frac{R_s + f(R_L)}{\omega L_s - \frac{1}{\omega C_s} + g(R_L)} \tag{12}$$

Then, according to the circuit principle, the transmitter impedance can be derived as:

$$Z_{sp} = R_p + j\left(\omega L_p - \frac{1}{\omega C_p}\right) + Z_r \tag{13}$$

Combining (11) and (13), the real part and imaginary part of $Z_r$ can be expressed as:

$$\begin{cases} real(Z_r) = real(Z_{sp}) - R_p \\ imag(Z_r) = imag(Z_{sp}) - \omega L_p + \frac{1}{\omega C_p} \end{cases} \tag{14}$$

Taking (14) into (12), the relationship between the transmitter impedance $Z_{sp}$ and the equivalent input impedances of the rectifier $R_e$ and $X_e$ can be established by the coefficient $\alpha$, as shown in (15).

$$\alpha = \frac{real(Z_{sp}) - R_p}{imag(Z_{sp}) - \omega L_p + \frac{1}{\omega C_p}} = -\frac{R_s + f(R_L)}{\omega L_s - \frac{1}{\omega C_s} + g(R_L)} \tag{15}$$

Taking (8) into (15), the relationship between $Z_{sp}$ and $R_L$ is obtained, as shown in (16).

$$nf(R_L) + mg(R_L) = -nR_s - m\left(\omega L_s - \frac{1}{\omega C_s}\right) \tag{16}$$

where

$$\begin{cases} m = real(Z_{sp}) - R_p \\ n = imag(Z_{sp}) - \omega L_p + \frac{1}{\omega C_p} \end{cases}$$

In (16), the circuit parameters such as self-inductance, internal resistance and compensation capacitance can be measured before identification, and there is no offset during the operation of the circuit basically. In addition, the real and imaginary parts of the transmitter impedance can be calculated by measuring the voltage on the parallel compensation capacitor at the transmitter $u_p$, the current of the transmitting coil $i_p$ and the phase difference

$\theta$ of the two, as shown in (17). Videlicet, the value of $R_L$ can be directly solved only by measuring the amplitude and phase of the voltage and current at the transmitter.

$$\begin{cases} real(Z_{sp}) = \dfrac{\dot{U}_p}{\dot{I}_p}\cos\theta \\ imag(Z_{sp}) = \dfrac{\dot{U}_p}{\dot{I}_p}\sin\theta \end{cases} \tag{17}$$

After solving the $R_L$, then the equivalent input impedance of the rectifier $R_e$ and $X_e$ can be calculated according to (8). Next, the real part or imaginary part of the reflected impedance $Z_r$ is calculated by (14), and then the mutual inductance $M$ can be obtained, as shown in (18).

$$M = \sqrt{\frac{[real(Z_{sp}) - R_p][R_s + f(R_L)]^2 + \left[\omega L_s - \frac{1}{\omega C_s} + g(R_L)\right]^2}{\omega^2[R_s + f(R_L)]}} \tag{18}$$

In conclusion, through the analysis of the real-imaginary ratio of the reflection impedance $Z_r$ and the calculation of the transmitter impedance $Z_{sp}$, the decoupling of the load and the mutual inductance in the identification process is cleverly realized. So far, the specific steps of joint identification of mutual inductance and load parameters of LCC-S IPT system have been described. From (16) and (18), it can be observed that the joint identification of mutual inductance and load parameters can be realized only by measuring the amplitude and phase of the voltage and current at the transmitter. The focus of the next step is how to obtain the functional relationship between the equivalent input impedance of the rectifier and the system load, that is, the explicit expression of Equation (8).

### 3.3. Modeling of Rectifier Equivalent Load Based on Data Fitting Method

A model of an IPT system as depicted in Figure 1 is built in the Matlab/Simulink environment, and the parameters of the model are shown in Table 1.

**Table 1.** Parametric Values of The IPT System.

| Parameters | Values | Parameters | Values |
|---|---|---|---|
| $U_{dc}$ (V) | 200 | $f$/kHz | 85 |
| $L_p$/μH | 128.9 | $L_s$/μH | 117.61 |
| $R_p$/Ω | 0.376 | $R_s$/Ω | 0.350 |
| $C_p$/nF | 32.58 | $C_s$/nF | 29.64 |
| $L_r$/μH | 25.2 | $M$/μH | 24.95 |
| $C_r$/nF | 136.8 | | |

The variation range of $R_L$ is set to 20~60 Ω, then the amplitude $A_u$ and phase $\varphi_u$ of the rectifier input voltage $u_{rec}$ and the amplitude $A_i$ and phase $\varphi_i$ of the rectifier input current $i_{rec}$ can be measured by Fourier module, respectively. Thus, the equivalent input impedance of the rectifier can be calculated by (19)

$$\begin{cases} R_e = \left|\dfrac{A_u}{A_i}\right| = \cos(\varphi_u - \varphi_i) \\ X_e = \left|\dfrac{A_u}{A_i}\right| = \sin(\varphi_u - \varphi_i) \end{cases} \tag{19}$$

In order to describe the variation rule between the equivalent input impedance of the rectifier and the system load accurately, the polynomial data fitting method is proposed to fit the data in this paper. Moreover, the coefficients of the polynomial are determined by the least square method. The specific steps are as follows: Firstly, the equivalent resistance $R_e$ and the equivalent inductance $L_e$ calculated by (19) under different load conditions are drawn and plotted in the same coordinate diagram. Then, the calculated data $R_e$ and $L_e$ are fitted by the first-order polynomial and second-order polynomial, respectively. Finally, the

data fitting diagram shown in Figure 7 can be obtained, and the fitting result is shown in Table 2.

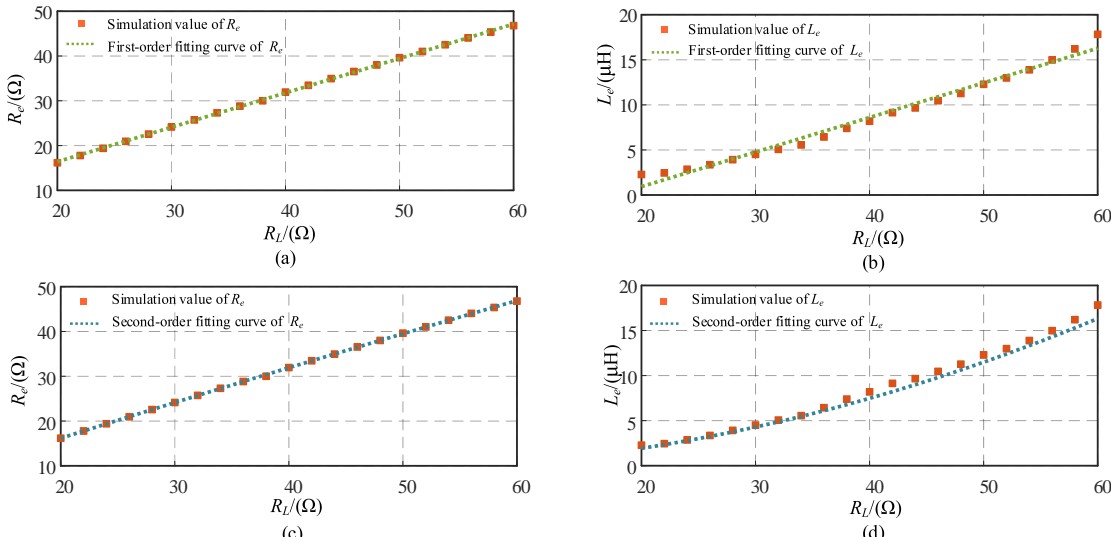

**Figure 7.** Fitting Curves of Equivalent Input Impedance of Rectifier: (**a**) First-order fitting curve of $R_e$; (**b**) first-order fitting curve of $L_e$; (**c**) second-order fitting curve of $R_e$; (**d**) second-order fitting curve of $L_e$.

**Table 2.** Data Fitting Results of Equivalent Impedance of Rectifier.

| Types of Polynomials | Data Fitting Equations | $R^2$/% |
|---|---|---|
| 1st-order polynomial | $R_e = 0.7683 + 1.042$ | 99.97 |
| | $L_e = 0.3838R_L - 6.746$ | 98.31 |
| 2nd-order polynomial | $R_e = 0.0009394\,R_L{}^2 + 0.8343R_L - 0.32316$ | 99.99 |
| | $L_e = 0.004416\,R_L{}^2 + 0.03056R_L - 0.3279$ | 99.83 |

As can be observed from Figure 7, the linearity between the rectifier equivalent input resistance $R_e$ and the system load resistance $R_L$ is high. Therefore, the fitting curves of first-order and second-order polynomials are basically consistent. As the system load resistance $R_L$ increases, the linearity of the rectifier equivalent input reactance $L_e$ gradually decreases. Therefore, the fitted curve is more consistent with the actual value when the order of the fitted curve is higher.

The data fitting results of the rectifier equivalent load are shown in Table 2. Usually, the goodness of fit is used to represent the fitting degree of the fitting curve to the actual value. In statistics, the statistic that measures the goodness of fit is the correlation coefficient $R^2$, which ranges from [0, 100]. If the value of $R^2$ is closer to 100, it means that the fitting effect of the fitting curve to the actual value is better. It can be observed from Table 2 that the higher the order of the polynomial is, the closer the value of $R^2$ is to 1. That is, the higher the goodness of fit of the data, the better the fitting effect.

When the first-order polynomial fitting is adopted, the functional relationship between the equivalent input impedance of rectifier $R_e$ and $X_e$ and the system load $R_L$ can be expressed as:

$$\begin{cases} R_e = a_1 R_L + b_1 \\ X_e = \omega L_e = a_2 R_L + b_2 \end{cases} \tag{20}$$

Combining (16) and (20), the relationship between $Z_{sp}$ and $R_L$ can be derived as:

$$R_L = -\frac{b_1 n + b_2 m}{a_1 n + a_2 m} \tag{21}$$

According to (16) and (17), $m$ and $n$ in (21) can be obtained by measuring the voltage on the parallel compensation capacitor at the transmitter $u_p$, the current of the transmitting coil $i_p$ and the phase difference $\theta$ between them, as shown in (22).

Where, $U_p$ is the amplitude of $u_p$ and $I_p$ is the amplitude of $i_p$.

Therefore, after measuring the physical quantities above, the load $R_L$ can be solved directly, and then the mutual inductance $M$ of the system can be identified.

$$
\begin{cases}
m = \dfrac{\dot{U_p}}{\dot{I_p}} cos\theta - R_p \\
n = \dfrac{\dot{U_p}}{\dot{I_p}} sin\theta - \omega L_p + \dfrac{1}{\omega C_p}
\end{cases}
\tag{22}
$$

Similarly, when the second-order polynomial fitting is adopted, the functional relationship between the equivalent input impedance of rectifier $R_e$ and $X_e$ and the system load $R_L$ can be expressed as:

$$
\begin{cases}
R_e = a_1 R_L{}^2 + b_1 R_L + c_1 \\
X_e = a_2 R_L{}^2 + b_2 R_L + c_2
\end{cases}
\tag{23}
$$

Therefore, the relationship between $Z_{sp}$ and $R_L$ can be derived as:

$$
(a_2 m + a_1 n) R_L{}^2 + (b_2 m + b_1 n) R_L + c_2 m + c_1 n = 0
\tag{24}
$$

In summary, it can be observed from (21) and (24) that $R_L$ can be solved by measuring the amplitude and phase of voltage on parallel compensating capacitor at the transmitter and current of transmitting coil. Moreover, the higher the order of the polynomial used for fitting is, the more complex the fitting formula about the equivalent input impedance of the rectifier module is. For example, $R_e$ and $X_e$ are quadratic functions of $R_L$ when fitted with a second-order polynomial. If used directly in the parameter identification process, it will increase the computational complexity greatly. Therefore, in order to facilitate the calculation and description, considering the goodness of data fitting and the simplicity of the parameter identification process comprehensively, this paper adopts the second-order polynomial fitting to analyze the equivalent load module of the rectifier. The identification process is shown in Figure 8.

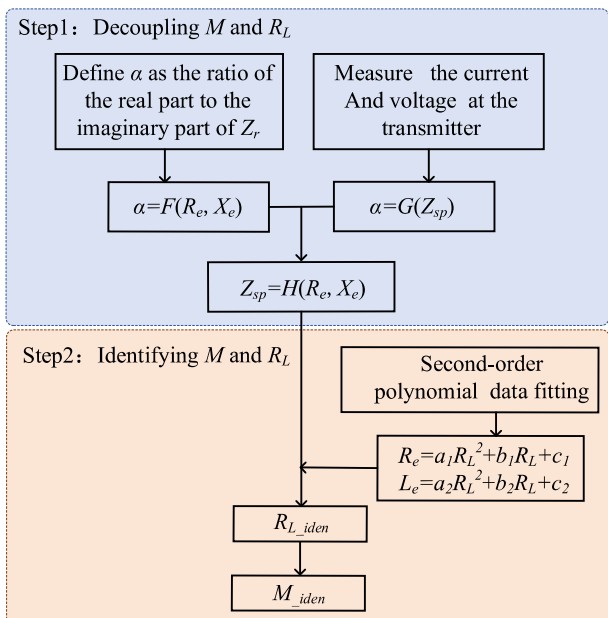

**Figure 8.** Flow chart of parameter joint identification method.

Observed from Figure 8, firstly, the functional relationship between the impedance of the transmitter and the equivalent input impedance of the rectifier can be established by defining the ratio of the real part to the imaginary part of the reflected impedance. Secondly, the functional relationship between the equivalent input impedance of the rectifier and the system resistance can be obtained by fitting the data with a first-order polynomial. Then, $M$ and $R_L$ can be solved by combining the two functional relationships above into an equation group. That is to say, the joint identification of mutual inductance and load parameters are realized under the condition that only the voltage and current at the transmitter are measured.

## 4. Simulation Verification and Discussion

In order to verify the feasibility and identification effect of the proposed identification method, an LCC-S IPT experimental platform is built, as shown in Figure 9, and the system parameters are shown in Table 1.

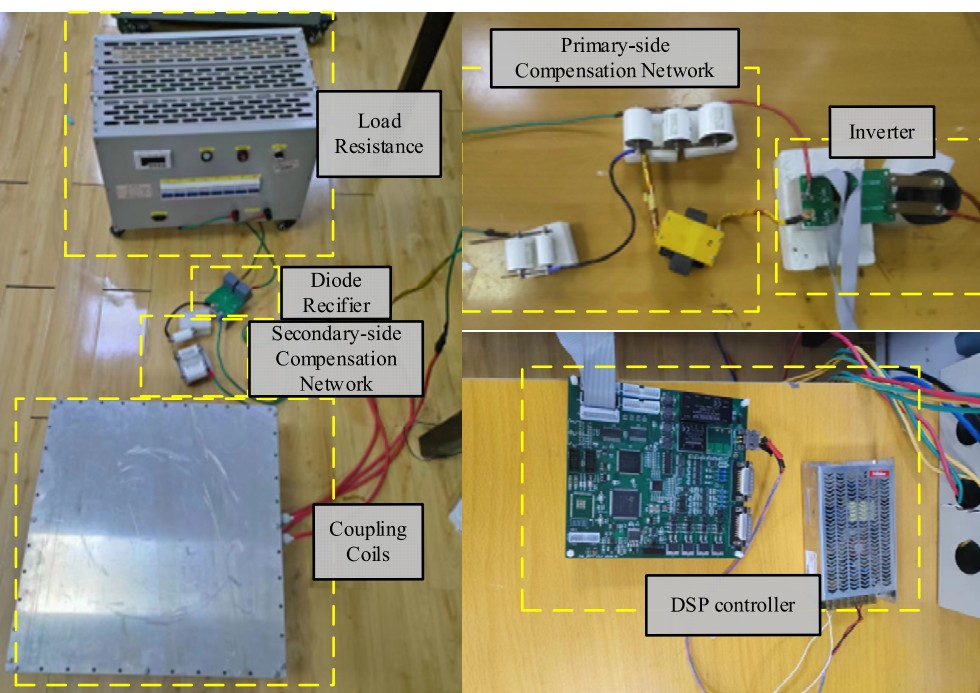

**Figure 9.** The experimental platform diagram of LCC-S IPT system.

The voltage and current waveforms of the rectifier of the LCC-S type IPT system under different load conditions are shown in Figure 10, where $u_{rec}$ is the rectifier input voltage and $i_{rec}$ is the rectifier input current. It can be observed that the waveform of the input current of the rectifier $i_{rec}$ has a certain distortion, which is not a standard sine wave, and its amplitude is to the right. This is consistent with the simulation analysis in Section 2. This can indicate that there is an inductance component in the rectifier load.

The waveforms of the voltage on the parallel compensation capacitor at the transmitter of the LCC topology $u_p$ and the current of the transmitting coil $i_p$ are shown in Figure 11. The experimental data of $u_p$ and $i_p$ are imported into Matlab for FFT (Fast Fourier transform) analysis, then the amplitude and phase of the fundamental waves of $u_p$ and $i_p$ are extracted, and then the parameters are identified.

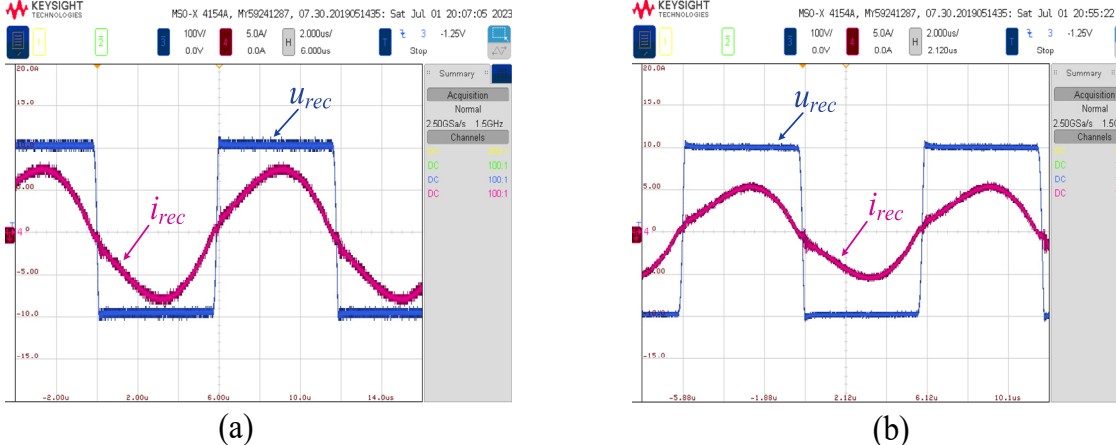

**Figure 10.** The voltage and current waveforms of the rectifier of the LCC-S IPT system: (**a**) $R_L$ = 40 Ω; (**b**) $R_L$ = 60 Ω.

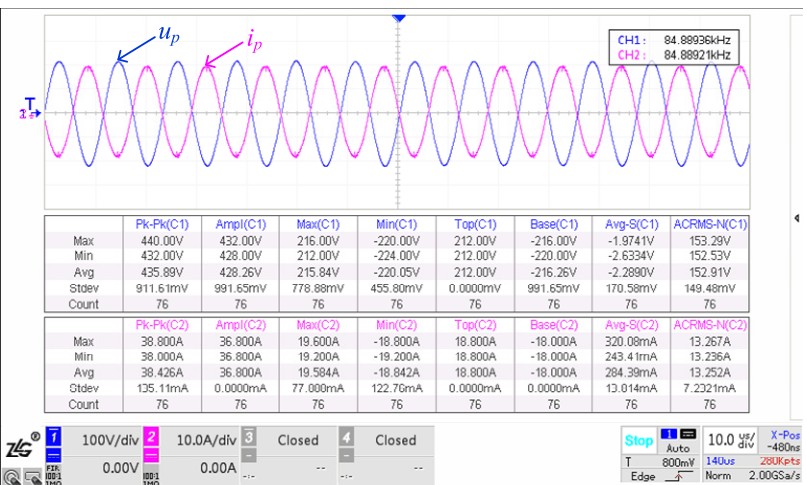

**Figure 11.** Voltage and Current Waveform of LCC-S Type IPT System when $M$ = 22.23 μH and $R_L$ = 40 Ω.

Compare the simulation value of $R_e$ and $L_e$ obtained in the Matlab/Simulink simulation with the data fitting value calculated by the second-order fitting Formula (22). It can be observed from Figure 12 that the maximum absolute error between the theoretical value and the fitted value of $R_e$ is 0.048 Ω and the maximum relative error of the two is 1.2%. Moreover, the maximum absolute error between the theoretical value and the fitted value of $L_e$ is 0.71 μH, and the maximum relative error of the two is 8.71%. When performing polynomial fitting on data, the higher the order of the polynomial used, the higher the fitting accuracy. Therefore, the errors above are mainly caused using second-order polynomial fitting in the data fitting process.

Figure 13 shows the identification results of $M$ and $R_L$ when the mutual inductance $M$ is set to 24.95 μH and the variation range of load $R_L$ is set from 25 Ω to 45 Ω with the size of 5 Ω per step. Compared with the theoretical value, the maximum relative error of the simulation identification result of the load $R_L$ is 4.71%, and the maximum relative error of the experimental identification result is 5.53%. The maximum relative error of the simulation identification result of the mutual inductance $M$ is 4.92%, and the maximum relative error of the experimental identification result is 5.20%.

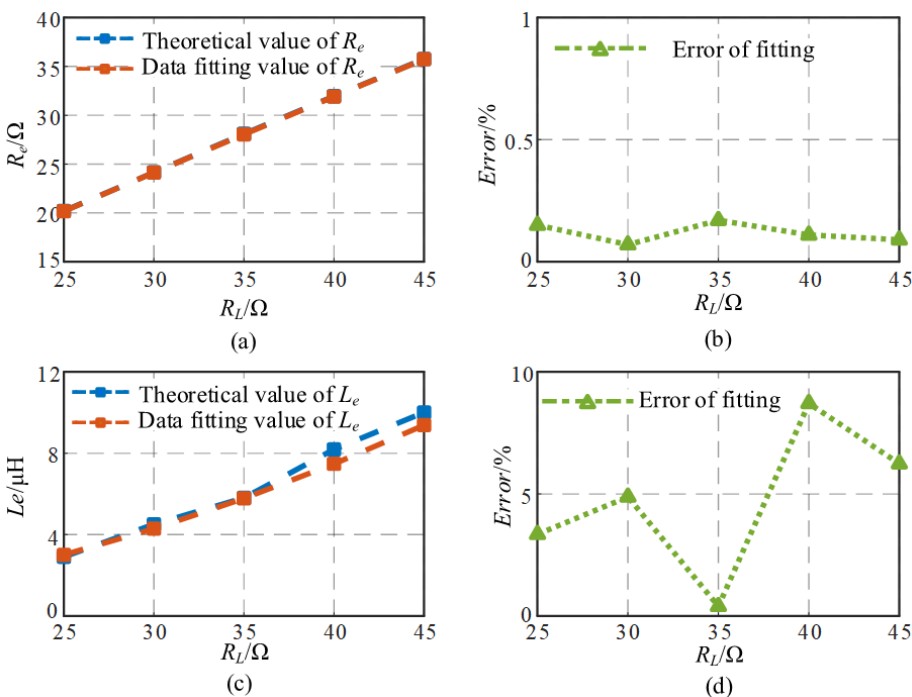

**Figure 12.** Simulation results of rectifier equivalent input impedance when $R_L$ changes: (**a**) $R_e$ data fitting results; (**b**) the error of data fitting of $R_e$; (**c**) $L_e$ data fitting results; (**d**) the error of data fitting of $L_e$.

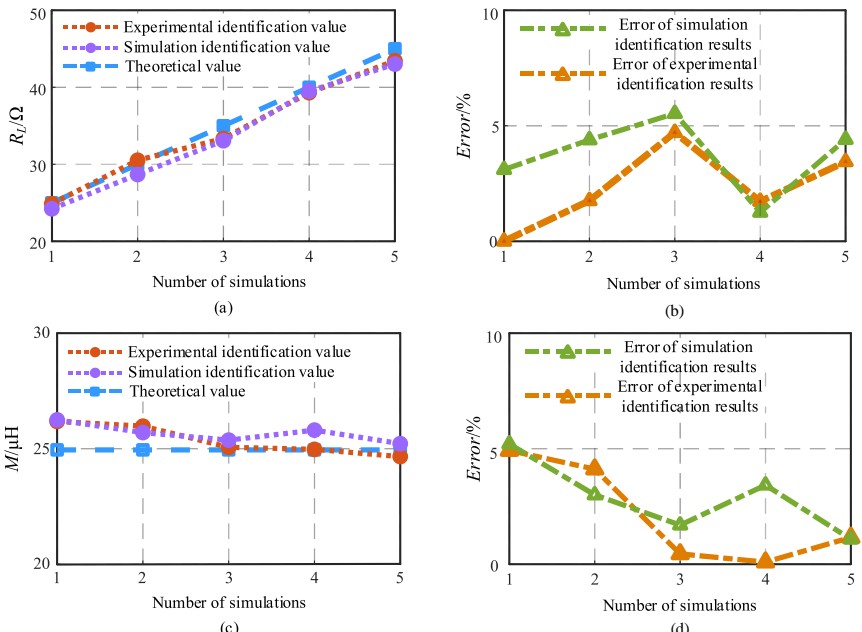

**Figure 13.** The identification results of mutual inductance and load when $M = 24.95$ μH. (**a**) Identification results of $R_L$; (**b**) the error of $R_L$ identification; (**c**) identification results of $M$; (**d**) the error of $M$ identification.

Figure 14 shows the identification results of $M$ and $R_L$ when the mutual inductance $M$ is set to 22.23 μH and the variation range of load $R_L$ is set to 25 Ω to 45 Ω with the size of 5 Ω per step. Compared with the theoretical value, the maximum relative error of the simulation identification result of the load $R_L$ is 2.96%, and the maximum relative error of the experimental identification result is 4.85%. The maximum relative error of the

simulation identification result of the mutual inductance *M* is 4.71%, and the maximum relative error of the experimental identification result is 3.33%.

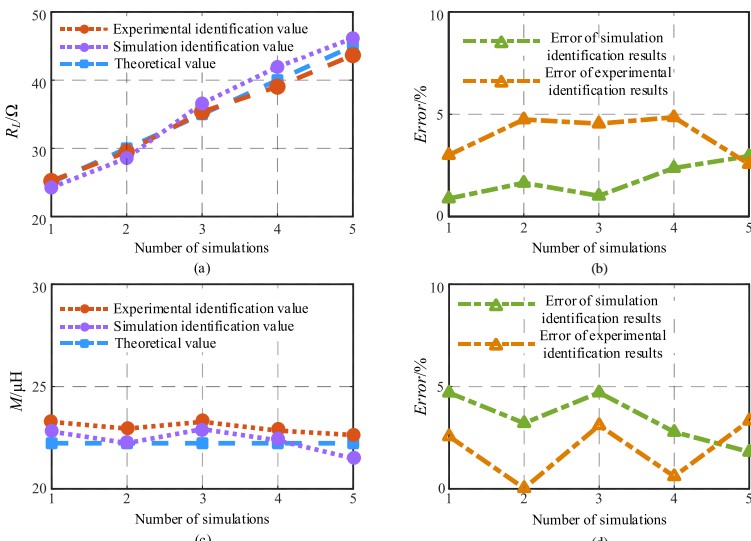

**Figure 14.** The identification results of mutual inductance and load when *M* = 22.23 μH. (**a**) Identification results of $R_L$; (**b**) the error of $R_L$ identification; (**c**) identification results of *M*; (**d**) the error of *M* identification.

By analyzing the results of the mutual inductance and load identification above, it can be known that the maximum absolute error of the mutual inductance *M* identification is 1.22 μH, and the maximum relative error is 5.2%. The maximum absolute error of load $R_L$ identification is 1.94 Ω, and the maximum relative error is 5.53%. Since the second-order polynomial fitting is used for the input impedance of the rectifier during the identification process, and this will obviously bring a certain identification error. However, these errors and differences are relatively small; thus, the universality and feasibility of the proposed identification method can be proved.

## 5. Conclusions

In this paper, a joint identification method of load and mutual inductance parameters is proposed for the LCC-S type IPT system considering the inductive component of a rectifier load. Firstly, the resistance–inductance characteristic of the rectifier load is revealed by simulation, and then the rectifier and the load are equivalent to a model in which a resistor and an inductor are connected in series. Secondly, by defining the ratio of the real part and the imaginary part of the system reflection impedance, the decoupling of mutual inductance and load is realized. Then, the functional relationship between the equivalent impedance of the rectifier and the system load resistance is obtained by data fitting. Finally, a joint identification of system load and mutual inductance parameters is realized by measuring the voltage and current of the transmitting side. In this paper, the resistance-inductance characteristics of the rectifier load of the LCC-S topology are considered, which improves the accuracy of the model. Moreover, on this basis, a parameter identification method is proposed, which can achieve high precision identification. A series of simulations and experiments are carried out to verify the proposed identification method. The load and mutual inductance identification results are in good agreement with the theoretical values, and the maximum errors are 5.53% and 5.2%, respectively, which proves the feasibility and accuracy of the identification method.

**Author Contributions:** Writing—original draft, H.S.; Writing—review & editing, X.W., P.S., L.W. and Y.L. All authors have read and agreed to the published version of the manuscript.

**Funding:** This work was funded by the National Natural Science Foundation of China under Grant 52007195 and in part Group Project in Hubei Province Natural Science Foundation of Innovation under Grant 2018CFA008.

**Data Availability Statement:** Not applicable.

**Conflicts of Interest:** The authors declare no conflict of interest.

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
