# Peer review of "Mutual Inductance and Load Identification of LCC-S IPT System Considering Equivalent Inductance of Rectifier Load"

_electronics, doi:10.3390/electronics12183841_

Round 1

Reviewer 1 Report

Thanks for the clear and detailed presentation. It is a well finished article. Below are some of the questions and suggestions:

1. This is a popular topic with numerous relevant studies. What is the benefit of the proposed method compared to the others (such as genetic or particle swarm algorithm)? Performance includes accuracy, applicable frequency and load resistance range are expected.

2. If the purpose of the identification without communication is to control the output voltge, then the comparison of computational resources among different methods should also be conducted. Will the close loop bandwidth large enough when facing a time-varing load?

3. Will wide-ranging variations in load affect the saturation level of inductors, consequently influencing mutual inductance? The papaer only shows the results from 900W-1600W. 

4. Line 138 139: Is that a typo in"current source for primary and voltage source for secondary" ? The secondary voltage is clamped by the DC output and then it should be a current source.

5. Line 177, there is a typo. It should be "inductive" rather than "resistive".

6. Figure 11 is too vague to be seen.

7. Where can I find the reference 16 and 17? They cannot be found in goolge scholar.

The overall English expression is commendable, with minimal errors. However, there is room for conciseness in the sections detailing the circuitry and diagrams.

Author Response

First, the authors would like to thank you for your efforts to review the manuscript and provide the feedback to make the presentation of this work better. The author has attached the responses in the attachment. Major changes are highlighted in the revised manuscript. As can be observed from the responses below, the authors did additional work, tried to provide detailed responses. Thank you again for all the efforts and useful important comments and suggestions.

Reviewer 2 Report

The work under consideration is a rigorous one that is in the field of current research on inductive power transfer technology.

Remarks:

1. in the title but also in the content, a series of abbreviations are used for which there is no corresponding explanation (eg: "LCC-S" line 11, 25, etc.; "SS" line 52)

2. Figure 3 has the name on page 5 and the image is on page 4 (see lines 148-149)

3. small drafting errors:

- add a space before the reference "[19]" line 73

- when writing equations, there is a differentiation between the positioning of the first three relationships compared to the rest (starting with relationship 4, the distance between the text and the mathematical relationship disappears).

- The same thing regarding the distance between the text and the content of the tables can be observed in their final part (see lines 268-270, respectively 293-295), compared to the beginning part;

- line 308 the symbol "~" between "(15) ~ (16)" should be replaced by "and";

- in the explanations to figures 13 and 14, line 380, respectively 391 "...(b) The error of M identification..." should be "(d)"

- in the entire bibliography the spacing should be checked (eg: "...Transmission[J]." line 430, it would be correct "...Transmission [J].")

4. sectorul dintre liniile 323 È™i 325 nu este formulat clar „...Mai mult, cu cât este mai mare ordinea polinomului utilizat pentru montare, cu atât formula de montare este mai complexă. impedanÈ›a de intrare echivalentă a modulului redresor este. . .."

5. nu pot fi găsite o serie de elemente bibliografice (ex: 2, 11, 16, 17, 21), ar fi util dacă ar putea fi verificate sau înlocuite

Author Response

(The authors gave the same response as above.)

Reviewer 3 Report

The paper is very interesitng. The aim of this paper is joint identification method of load and mutual inductance for LCC-S IPT system, which does not require the establishment of primary and secondary communication and related control.

In order to increase the quality of the paper, the following should be improved:

1. Figure 11 is illegible

2. In paper presents the formula how to calculate the mutual inductance M, this parameter depends also from e coupling coefficient. It should be more explainted.

3. The conclusion should be corrected with an indication of what was improved and what is new in this paper. It's too modest in its current form.

4. References - to small number of references position it is worth add some position for example

Detka, K.; Górecki, K. Wireless Power Transfer—A Review. Energies 2022, 15, 7236. https://doi.org/10.3390/en15197236

Shan, D.; Wang, H.; Cao, K.; Zhang, J. Wireless power transfer system with enhanced efficiency by using frequency reconfigurable metamaterial. Sci. Rep. 2022, 12, 331

Geng, Y.; Sun, H.; Yang, Z.; Li, B.; Lin, F. A High Efficiency Charging Strategy for a Supercapacitor Using a Wireless Power Transfer System Based on Inductor/Capacitor/Capacitor (LCC) Compensation Topology. Energies 2017, 10, 135

Author Response

(The authors gave the same response as above.)
